# Testing, tracing and isolation in compartmental models

**Simone Sturniolo**[1], **William Waites**[2], **Tim Colbourn**[3], **David Manheim**[4], **Jasmina Panovska-Griffiths**[3,5,6]*

**1** Scientific Computing Department, UKRI, Rutherford Appleton Laboratory, Harwell, United Kingdom, **2** School of Informatics, University of Edinburgh, Edinburgh, United Kingdom, **3** UCL Institute for Global Health, London, United Kingdom, **4** University of Haifa Health and Risk Communication Research Center, Haifa, Israel, **5** Department of Applied Health Research, UCL, London, United Kingdom, **6** Wolfson Centre for Mathematical Biology and The Queen's College, Oxford University, Oxford, United Kingdom

☉ These authors contributed equally to this work.
¤ Current address: Centre for the Mathematical Modelling of Infectious Diseases, London School of Hygiene and Tropical Medicine, London, UK is current address for W.Waites
* j.panovska-griffiths@ucl.ac.uk

**Data Availability Statement:** Data underlying the findings are available at https://github.com/ptti/ptti/tree/ptti-theory-paper.

**Funding:** WW was supported by the Chief Scientist Office Scotland (COV/EDI/20/12). JPG was

## Abstract

Existing compartmental mathematical modelling methods for epidemics, such as SEIR models, cannot accurately represent effects of contact tracing. This makes them inappropriate for evaluating testing and contact tracing strategies to contain an outbreak. An alternative used in practice is the application of agent- or individual-based models (ABM). However ABMs are complex, less well-understood and much more computationally expensive. This paper presents a new method for accurately including the effects of Testing, contact-Tracing and Isolation (TTI) strategies in standard compartmental models. We derive our method using a careful probabilistic argument to show how contact tracing at the individual level is reflected in aggregate on the population level. We show that the resultant SEIR-TTI model accurately approximates the behaviour of a mechanistic agent-based model at far less computational cost. The computational efficiency is such that it can be easily and cheaply used for exploratory modelling to quantify the required levels of testing and tracing, alone and with other interventions, to assist adaptive planning for managing disease outbreaks.

## Author summary

The importance of modeling to inform and support decision making is widely acknowledged. Understanding how to enhance contact tracing as part of the Testing-Tracing-Isolation (TTI) strategy for mitigation of COVID is a key public policy questions. Our work develops the SEIR-TTI model as an extension of the classic Susceptible, Exposed, Infected and Recovered (SEIR) model to include tracing of contacts of people exposed to and infectious with COVID-19. We use probabilistic argument to derive contact tracing rates within a compartmental model as aggregates of contact tracing at an individual level. Our adaptation is applicable across compartmental models for infectious diseases spread. We show that our novel SEIR-TTI model can accurately approximate the behaviour of

supported by the National Institute for Health Research (NIHR) Applied Health Research and Care North Thames at Bart's Health NHS Trust (NIHR ARC North Thames). The funders had no role in study design, data collection, data analysis, data interpretation, or writing of the report. The views expressed in this article are those of the authors and not necessarily those of the NHS, the NIHR, or the Department of Health and Social Care.

**Competing interests:** The authors have declared that no competing interests exist.

mechanistic agent-based models at far less computational cost. The SEIR-TTI model represents an important addition to the theoretical methodology of modelling infectious disease spread and we anticipate that it will be immediately applicable to the management of the COVID-19 pandemic.

This is a *PLOS Computational Biology* Methods paper.

## Introduction

Since the beginning of 2020, the W orld has been in the midst of a COVID-19 pandemic, caused by the novel coronavirus SARS-CoV-2. To slow down the spread, many countries, including the UK have imposed social distancing mitigation strategies. However, such measures cannot feasibly be imposed over a long period as this may lead to economic collapse. As a consequence countries need to consider how to ease lockdown measures while controlling SARS-CoV-2 spread.

The World Health Organisation has recently updated their guidance on this, recommending a six point strategy that requires firstly assuring that the pandemic spread has been suppressed, and is followed by detecting, testing, isolating and contact-tracing of infected individuals [1].

Mathematical modelling has figured prominently in decision making around control and containment of COVID-19 spread, including the imposition of physical distancing measures [2]. It provides a logical framework for understanding the propagation of an infectious disease through a population and allows different interventions to be explored, including testing and contact tracing of infected individuals as possible strategies to ease social distancing restrictions. Such models are also necessarily simplifications and understanding of their assumptions and what they do and do not represent is required to correctly interpret them.

Mathematical models have a long history of being used to describe the spread of infectious diseases from plague outbreaks more than a century ago [3] to the more recent SARS [4] and Ebola [5], [6] epidemics, and from making decisions around different vaccination strategies for influenza [7, 8] to modelling HIV [9, 10], and from modelling pandemic influenza [11] to currently facilitating real-time policy decision making around the COVID-19 epidemic [12–19]. There are several common approaches, each with advantages and disadvantages [20, 21]. Compartmental models [21–23] partition the population into different compartments such as susceptible, exposed to the virus but not infectious, infectious and removed and track the movements of individuals between these groups. Though dynamics of real disease outbreaks are fundamentally stochastic [24–26], this level detail is mainly relevant for early stages or small outbreaks [27]. Commonly within compartmental models a mean-field approximation given by ordinary differential equations (ODE) is used [21, 28, 29]. The latter approach is particularly attractive because it is computationally efficient and can yield informative results. ODE systems can be generalised to explicitly incorporate dependence on system state at some times in the past, yielding delay-differential equations (DDE) [30–32], the analogue for continuous state of Markov processes with finite memory. Such formulations require meticulous care to solve accurately [33, 34] and much of what is known about their behaviour consists of asymptotic results [35–38]. Branching processes are used [14, 29, 39, 40] where more flexibility is desired in representing the timing of transitions among compartments and, for continuous time, are amenable to stochastic differential equation (SDE) treatment. For some choices of distribution, the SDE formulation is Markovian and can be analysed as a continuous-time

Markov chain (CTMC) [25, 41]. Finally, individual- or agent-based models (IBM/ABM) explicitly represent each individual in the population and allow for fine-grained modelling of the characteristics of each one such as different contact patterns or susceptibilities to the disease [42–46]. They have been [47], and are being [15–17] widely used for planning and epidemic control. While ABMs allow for maximal flexibility and realism, this comes at a high computational cost and it can be difficult to extract analytical results that relate the fine-grained behaviour to population-level effects. It is generally feasible to conduct agent-based simulations for populations of tens of thousands, but there are salient features of epidemics such as the timing and size of peaks of infectious individuals that depend on population sizes two orders of magnitude larger. An important subset of ABMs are network or graphical models [48–53] where the structure of the population, the possible interactions among its members, are explicitly represented. In addition to the computational cost and analytical difficulties with ABMs, sufficient data to support their fine-grained realism is rarely available. For many purposes, including the one that we are concerned with here, an accurate qualitative understanding of the effect of interventions like testing and contact tracing, cheap, coarse, high-level models are more useful than expensive fine-grained models that rely on vast often not readily available data.

While classic compartmental models can easily be used to simulate some interventions analogous to parameter changes, they cannot readily include contact tracing of infected individuals unless vast assumptions are made. This is because modelling contact-tracing is intrinsically reliant on individual behaviour within a network structure. Previous work on Ebola [6], SARS [4] and COVID-19 used simple approaches to represent contact tracing in a compartmental model: asserting that a constant fraction of exposed individuals becomes isolated due to contact tracing [15, 19, 54, 55] or reducing transmission by a constant amount, perhaps after a delay [56]. We believe that existing approaches are insufficient for the purpose of understanding how the rate and timing of testing and contact tracing affect success in containing outbreaks. The purpose of contact tracing is to attempt to isolate infectious, or soon to be infectious individuals. Although attempts have been made to model contact tracing in combination with isolation [57], the two processes were modelled independently, deriving an analytical solution that overestimated the tracing function. Furthermore, contact tracing should result in the isolation of both infectious and exposed individuals and this is a key assumption that previous work has missed. Contact tracing will also inevitably result in the isolation of susceptible and recovered individuals with the former contributing to a reduced rate of disease propagation. To properly understand this process it is imperative to model the effects of contact tracing with mathematical rigour.

In this paper we develop an extension to the classic Susceptible-Exposed-Infectious-Removed (SEIR) model [22, 58, 59] simulated with ODEs to include testing, contact-tracing, and isolation (TTI) strategies. We call this model SEIR-TTI. This model captures the salient features of the manifestation at the population level of the dynamics of testing and tracing at the individual level.

We note that slightly different nomenclature for SEIR has ben used by different authors. Exposed means infected but not yet infectious and is sometimes called Latent. Infectious is sometimes called Infective and represents individuals capable of transmitting the disease. Removed is often called Recovered, though we opt for the former as it indicates that those individuals are no longer causing infection but we make no statement about whether they are removed through recovery or death.

Due to its relative simplicity, SEIR-TTI is applicable across a spectrum of diseases. With appropriate parametrisation, it can be used anywhere a standard SEIR model can be used with the same caveats and limitations.

Though we are clearly motivated by the current COVID-19 pandemic and wish to understand how interventions like TTI can be used to contain it, we do not claim that we are modelling it in particular. Our contribution is a mathematical tool and software implementation that can be used for understanding TTI, not a model of COVID-19.

The method that we present is general and can also be applied to other compartmental models, with the standard caveat that with more compartments comes more work to determine the appropriate rates that need to be informed by data. We validate our SEIR-TTI ODE model against a mechanistic agent-based model where testing, tracing and isolation of individuals is explicitly represented and show that we can achieve good agreement at far less computational cost. We also provide a flexible software package at https://github.com/ptti/ptti/tree/ptti-theory-paper with a convenient declarative language for specifying parameters and interventions and implementations of the SEIR-TTI ODE model, mechanistic agent-based model, a second non-mechanistic rule-based model in the $\kappa$-language formalism [60, 61], and several related models such as classic SEIR.

## Results

We design a compartmentalised model describing the populations of susceptible ($S$), exposed ($E$—infected but not infectious), infectious ($I$) and removed ($R$) population cohorts.

These models are widely used to describe the spread of various infectious diseases with disease progression captured by movement of individuals sequentially between compartments accounting for progression from susceptible individuals ($S$) being exposed to the virus and becoming infected but not infectious ($E$), to becoming infectious ($I$) until they recover ($R$). A schematic illustrating this model is shown in Fig 1.

The novelty of our model is that we have within each compartment included subgroups of people diagnosed and undiagnosed with the virus, attributable to reported and unreported diagnosis. Individuals in our model are defined to be diagnosed either through testing or putatively through tracing. Diagnosed individuals are then isolated.

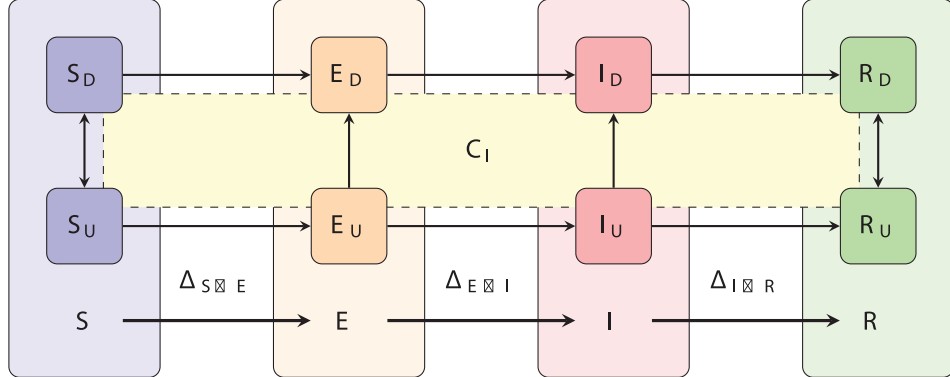

**Fig 1. Schematic of an SEIR model with diagnosis described by testing and contact-tracing.** SEIR is a compartmentalised model describing susceptible ($S$), exposed ($E$—infected but not infectious), infectious ($I$) and removed ($R$) population cohorts. Individuals move between these compartments in sequence as they become exposed, infected and infectious during disease progression until recovery. The novelty here is that each compartment comprises diagnosed and undiagnosed individuals with diagnosis leading to isolation. We assume that diagnosis happens through testing or putatively through tracing. Tracing is mediated through contact, and the intersection with $C_I$ represents contact with an infectious individual. Non-infectious individuals having been isolated through contact tracing have, in effect, been misdiagnosed. Individuals transition between compartments $X$ and $Y$ at rates $\Delta_{X \rightarrow Y}$ which we derive in the text.

## The effect of testing and isolation alone

Before introducing contact tracing, we examine the standard SEIR model with testing. These results, and those in the following section, use the system of differential equations as described in detail in the Methods. We choose a relatively large initial number of infectious individuals merely for illustrative purposes as it renders the dynamics clearer—the more aggressive testing regimes would result in immediate containment of a small outbreak which would be difficult to see whereas a large outbreak nevertheless takes some time to contain. The parameters have the usual meaning, with values fixed for the purposes of this section: $N = 6.7 \times 10^7$ individuals is the total population, $I(0) = 10^5$ is the initial number of infected individuals, $\hat{\beta} = 0.033$ infections/contact is the probability of transmission; $c = 13$ contacts/day is the contact rate, $\alpha = 0.2$ days$^{-1}$ is the incubation rate, the rate of leaving the exposed state and becoming infectious; and $\gamma = 7^{-1}$ days$^{-1}$ is the rate of recovery, or leaving the infectious state. These values result in a basic reproduction number of $R_0 = 3$. In the simplest case, testing is conducted at random at some rate $\theta$ of tests per infectious individual per day and those that receive a positive result are immediately isolated.

Representative trajectories from this system for various values of $\theta$ are shown in Fig 2. The upper panel shows the time-series for total infections, exposed and infectious, and the lower panel shows the effective reproductive number, $R_e(t)$. We can observe that while testing the entire population every 20 days ($\theta = 0.05$) results in a lower maximum total number of infections, we require very frequent testing, every 3-4 days ($\theta = 0.3, 0.25$) in order to control an outbreak and cross the $R_e(t) = 1$ threshold (red horizontal line). It is straightforward to work out the condition under which testing crosses this threshold by analysing the fixed points in the underlying system of differential equations since the required condition is that there is no change in the number of infectious people as they each infect one other on average and then are removed. Some arithmetic yields $\theta_{\mathrm{crit}} = \hat{\beta} c - \gamma$, the red line in Fig 3.

The above shows that, whilst testing and isolating alone can be sufficient to control an outbreak, it would take a herculean effort on its own. Without any form of distancing ($c \approx 13$) it is necessary to conduct tests about every 3.5 days. If a sizeable number of infected individuals are asymptomatic, there is no alternative but to test the entire population at this rate. Imposing strict social distancing measures can help. If contact rate is cut by half, the required rate is closer to once per fortnight. There is, however, a strategy to avoid regularly sampling the entire population in order to direct tests to those most likely to be infected: contact tracing, which we consider next.

## The effect of contact tracing

The central mathematical result is the expression for the rate at which individuals are isolated due to contact tracing,

$$\Delta_{X_U \to X_D}^{(C_T)} = \eta \, \theta \chi \Pr(C_I | X_U) X_U \tag{1}$$

where $\eta$ and $\chi$ are the probability of success and the rate of contact tracing respectively and $\theta$ is the rate of testing as before. The notation is explained in detail in the methods section, but the intuition is that, for any compartment $X$, divided into unconfined, $X_U$, and isolated, $X_D$, sub-compartments, the rate of moving between them is proportional to the probability of having had contact with an infectious individual conditional on being in $X_U$.

The effects of contact tracing is shown in Fig 4. The scenario is the same as with testing alone, except that the testing rate is fixed at $\theta = 14^{-1}$days$^{-1}$ and the tracing rate is fixed at $\chi = 2^{-1}$days$^{-1}$. The tracing success rate, $\eta$, is allowed to vary. The interpretation is that, on average,

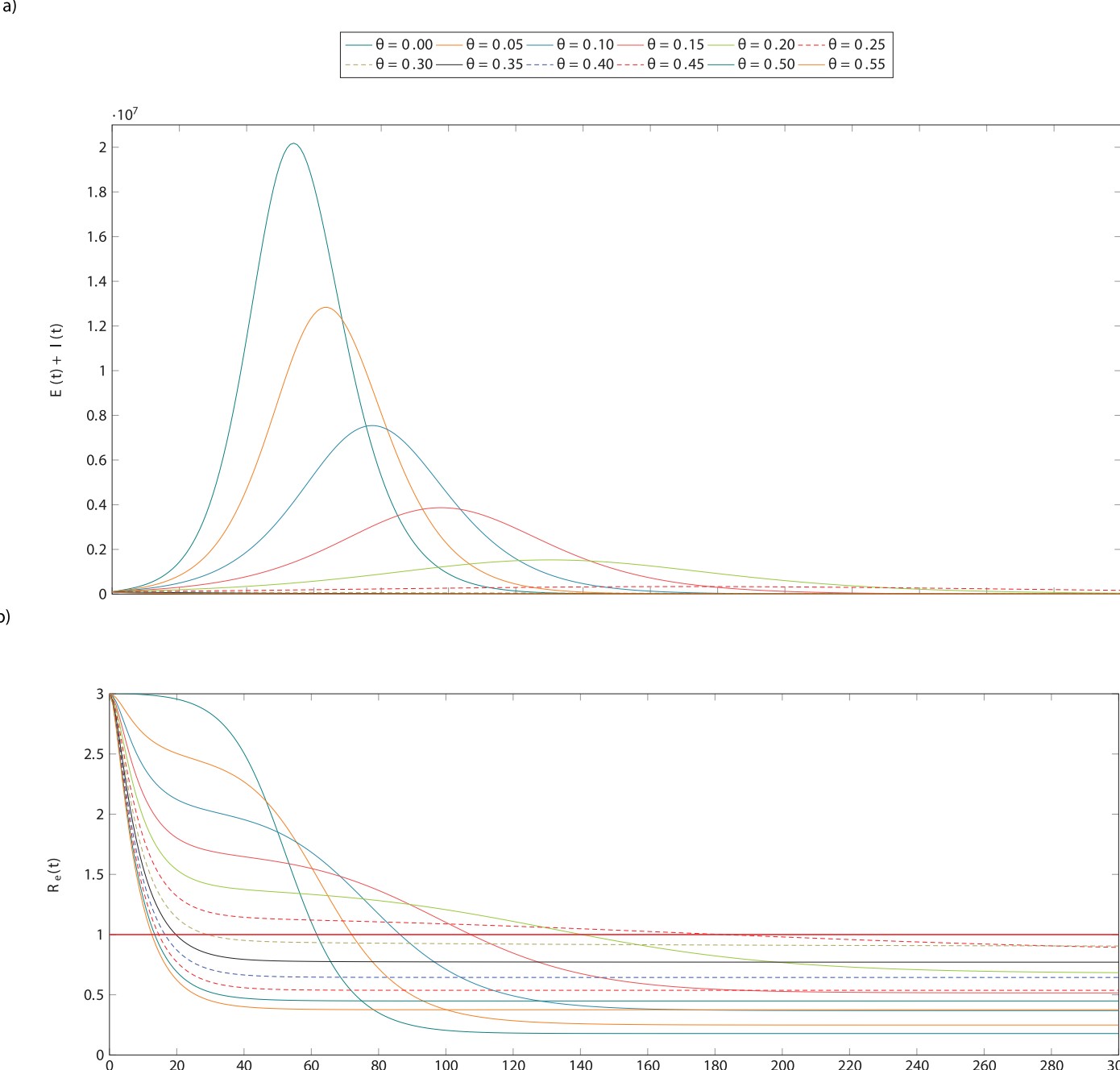

**Fig 2. The effect of testing and isolation alone in a hypothetical population.** The dynamics represented here are for a scenario with normal contact, $c = 13$, and an initial number of infected individuals, $I(0) = 100,000$. Individuals who test positive are isolated for the duration of their illness. The top plot shows the total infections (exposed and infectious individuals) over time for various testing rates ranging from none, $\theta = 0$, to testing all infectious individuals every two days, $\theta = 0.55$. The bottom plot shows the reproduction number over time for these same scenarios. Observe that even fairly frequent testing, e.g every five days, $\theta = 0.2$, this is only sufficient to reduce peak infections by one order of magnitude from about 20 million to about two million. In the infrequent testing regimes, $\theta \in [0.05, 0.25]$, we can also observe that the curve described by $R_e(t)R(t)$ is not a sigmoid but instead first falls to a value above $R(t) = 1$ before stabilising and then falling again. This is because though testing and isolating does have an effect at those rates, it is not sufficiently frequent to identify all of those who are infectious.

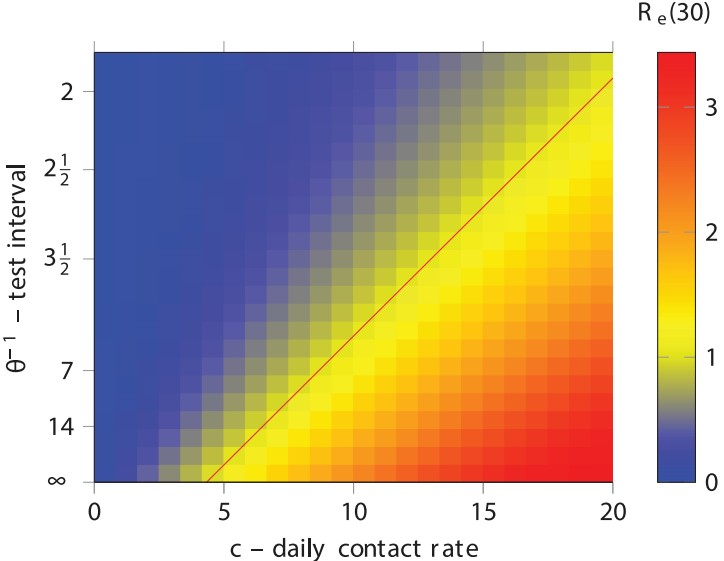

**Fig 3. Reproduction number after 30 days for various values of the contact rate, $c$ and the testing rate, $\theta$.** The red line is is given by the equation $\theta_{\mathrm{crit}} = \hat{\beta} c - \gamma$. As above, $\hat{\beta} = 0.033$ and $\gamma = 0.1429$.

an infectious individual expects to be tested in 147 days and contacts can expect to be traced in 2 days. The choice of these values for illustrative purposes is purposeful. Recall from the previous section that $\gamma$, the recovery rate is fixed at $7^{-1}\mathrm{days}^{-1}$. One would expect that testing and isolating individuals, on average, after they have recovered and it is too late would be insufficient to contain an outbreak. Indeed it is not sufficient, but it does reduce the maximum number of infected individuals somewhat. However, since tracing happens as a consequence of testing, it amplifies its effectiveness. This can be seen in the figure where even a modest tracing success rate of 30-40% results in a substantial reduction of more than half the peak infections.

The relationship between testing rate and tracing rate can be seen from Fig 5. When $\theta$ is very small, meaning very little testing, then contact tracing has little effect. This is unsurprising because testing causes tracing. When there is very frequent testing, on the other hand, there is little benefit to contact tracing. When testing happens more frequently on average than an individual can infect another, it is sufficient to control the outbreak on its own. However for intermediate values, contact tracing amplifies the effectiveness of testing. The above result can be seen from this plot as well: when testing of infectious individuals is expected in a week, a modest 40% success rate at tracing contacts in two days is enough to reduce the reproduction number from 2 to less than 1.5, a substantial benefit.

## Ordinary differential equations and agent-based models

The central result of this paper is not specific observations about how testing and contact tracing affect the propagation of epidemics, though those are valuable, but a technique to compute these effects efficiently. This technique allows consideration of larger populations than would be possible with agent- or individual-based models allowing for the exploration of many different scenarios. Figs 3 and 5, for example, each contain $25 \times 25 = 525$ data points resulting from a separate simulation. Performing these 1050 total simulations takes under a minute on a regular laptop. This would have not been possible with agent- or individual-based models, with population sizes in the hundreds of thousands or millions.

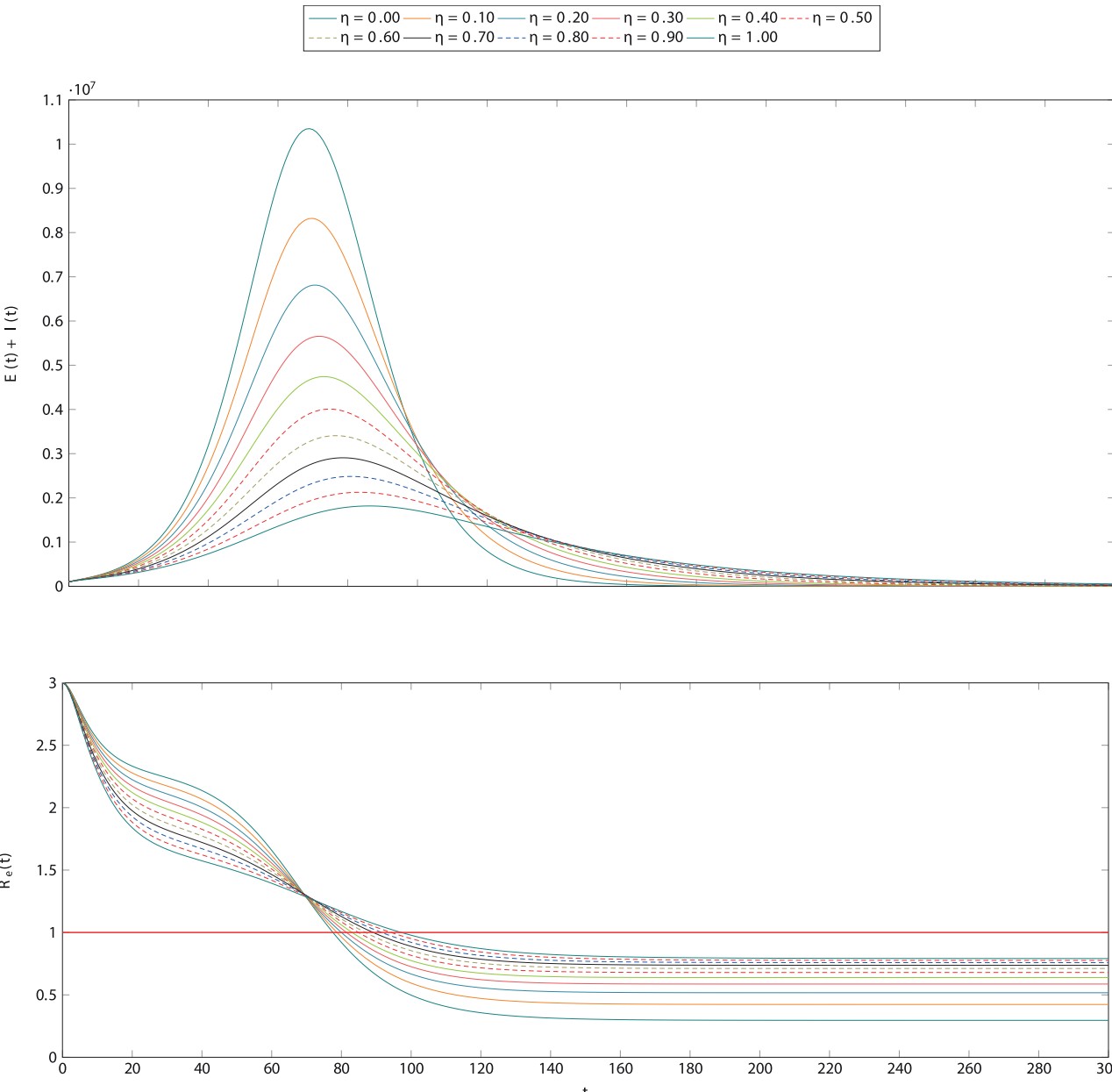

**Fig 4. The effect of testing, tracing in a hypothetical population.** The dynamics presented here are the same as those of Fig 2 with a testing rate $\theta = 14^{-1} \text{days}^{-1}$, meaning testing of infectious individuals on average once per two weeks once per week. The rate of contact tracing is set at $\chi = 2^{-1} \text{days}^{-1}$, meaning that it takes on average two days to trace a contact. A variety of values of tracing success rate, $\eta$ are explored. Under these conditions, even a modest success rate of 30-40% enhances testing and results in a maximum of infectious individuals that is about half the magnitude with testing alone. The lower panel is, as above, the corresponding time-series for the reproduction number.

It could be argued that it is sufficient to capture these dynamics in an agent-based model for modest populations and simply rescale the output for large populations. That approach is not sound for two reasons that are easily seen. First, small outbreaks. Imagine a hypothetical country of 70 million people with 100,000 infections. Proportionally, that is 14.3 infections in a population of 10,000. There is a non-negligible probability that an outbreak of size 14 will die out on its own. This will be accounted for by the ABM but is not a realistic possibility for an

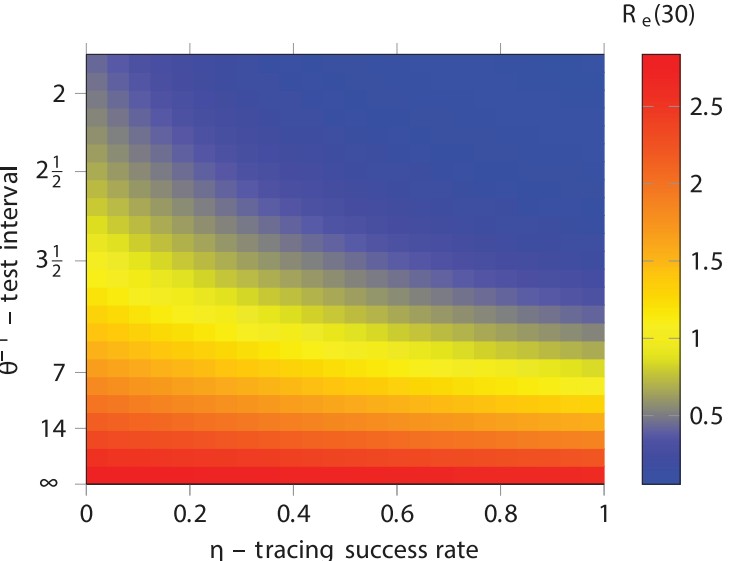

**Fig 5. Reproduction number after 30 days for various values of the testing rate, $\theta$ and the contact tracing success $\eta$.**

outbreak of 100 thousand. Scaling therefore suggests fundamentally different results. Second, without intervention, the number of infectious individuals will reach a maximum as the available pool of susceptible individuals becomes depleted. This takes longer in a large population simply because the pool is larger. If timing of the peak of an outbreak is a quantity of interest, a scaled ABM will give the wrong result.

However, doing this requires some approximations and it is important to understand where and how well these approximations hold. To do this, we compare with two different agent based models as described in the methods, and show that our method agrees well for a large range of physically interesting and realistic parameter values. The first ABM reproduces the same uncorrelated processes as the ODEs, with agents moving between compartments at constant rates, without any correlation with their time of arrival in them. This results in an exponential distribution of the times that each agent spends in a given state. However, in reality, the distribution of these times is rarely exponential; more realistic choices are distributions with a maximum at $t > 0$ [62–66]. Therefore we also try a second correlated ABM, in which agents all stay inside each compartment for a fixed amount of time, after which they transition. This can be seen, mathematically, as the permanence times having a Dirac distribution instead. All compartments and rules for this model are the same, and the rates are picked so that the time for each transition equals the average time for the exponential distribution in the other models. More details are provided in the methods section.

A comparison of the ODE and the first type of ABMtwo systems for reasonable parameter values is shown in Fig 6. The figure shows good agreement between the mean trajectory of the ABM and the ODE approximation. The agreement is particularly precise for the exposed and infectious compartment of both varieties. We can observe a slight over-estimate of the number of unconfined susceptible individual and corresponding under-estimate of the unconfined removed ones. These over- and under-estimates are nevertheless acceptably close with a relative error in the magnitude of the susceptible population of under 10%.

There exist extreme scenarios where the ODE performs poorly at reproducing the mean trajectory of the ABM system. An example is shown in Fig 7. One such scenario is when the

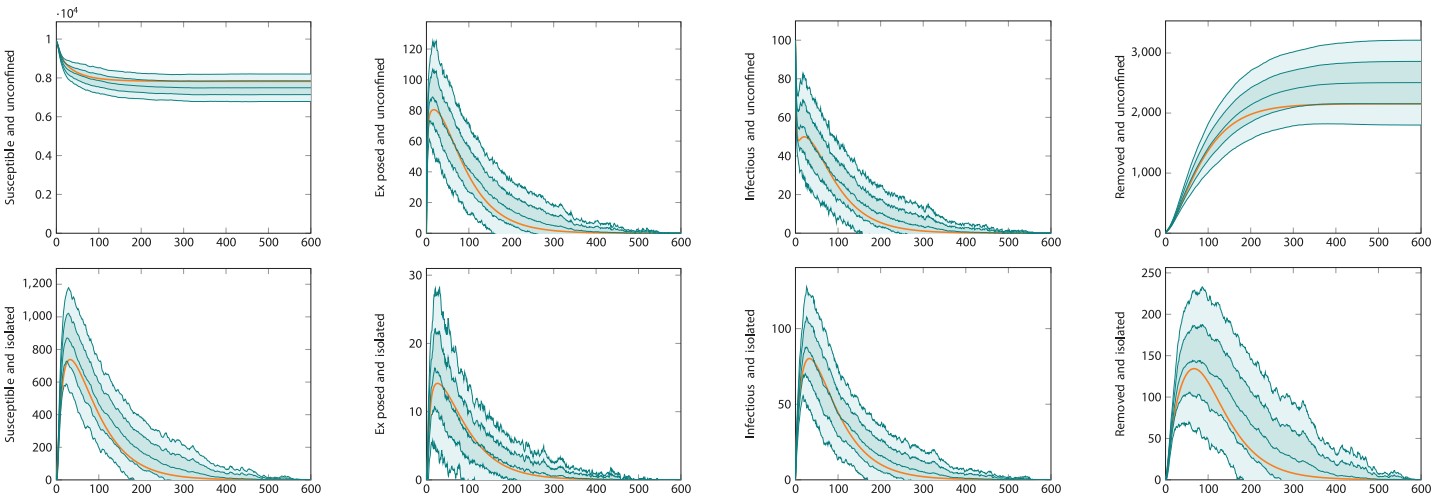

**Fig 6. Comparison of differential equation and agent-based model.** Here the population size is 10000 individuals with 100 initially infected. The testing rate is $\theta = 7^{-1}\mathrm{days}^{-1}$, and tracing rate and success probability are $\chi = 0.5$, $\eta = 0.5$. The plots show the time-series for each compartment. The top row are the compartments representing unconfined individuals, those in $S_U$, $E_U$, $I_U$ and $R_U$. The bottom row are those representing isolated—diagnosed or distanced—individuals, $S_D$, $E_D$, $I_D$, $R_D$. The heavy orange curves are the output of the ODE-based simulation. The teal curves are the average output of the agent-based simulation, with envelopes for one and two standard deviations.

testing rate is very low. The figure shows when $\theta = 50^{-1}\mathrm{days}^{-1}$. This circumstance violates the assumption underlying Eq 22 that the number of susceptible contacts available for tracing should be much smaller than the total susceptible population. Intuitively, this can be understood as the ODE approximation holding well when testing and tracing are conducted sufficiently rapidly to perform their required purpose. When they do not, the approximation is poor. Even in this extreme scenario, however, where the curve produced by the ODE system is several standard deviations distant from the average trajectory of the ABM, its shape is still similar and realistic.

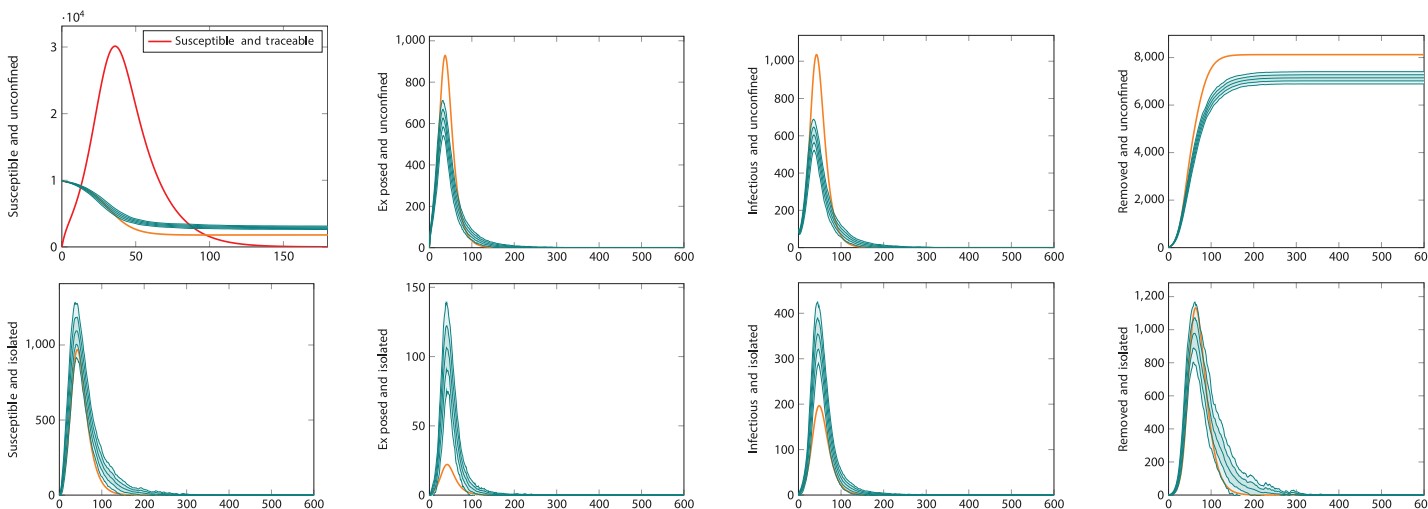

**Fig 7. Pathological parameter values.** This plot shows the effect of very low levels of testing, $\theta = 50^{-1}\mathrm{days}^{-1}$. In this circumstance, the number of traceable susceptible individuals takes on unphysically high values, shown by the red line in the top left panel. This results in an overestimation of the maximum number of unconfined exposed and infectious individuals and a corresponding underestimation of the effect of contact tracing in preventing infection in this scenario.

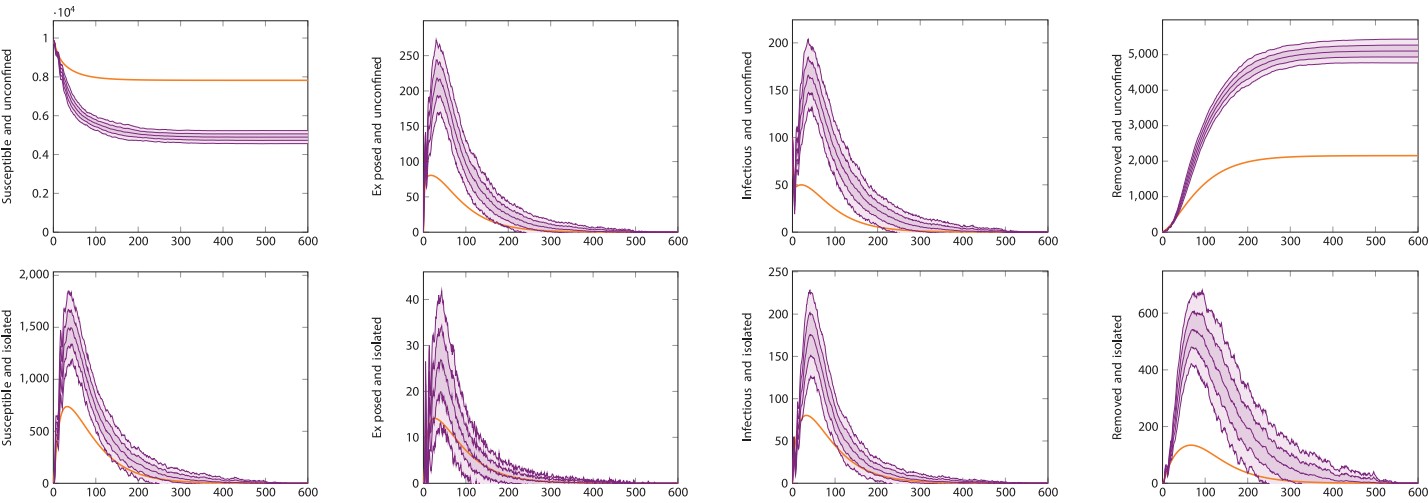

**Fig 8. Comparison of differential equation and agent-based model with correlated transitions.** Here the population size is 10000 individuals with 100 initially infected. The testing rate is $\theta = 7^{-1}$days$^{-1}$, the base testing rate is $\theta_0 = 2\theta$, and tracing rate and success probability are $\chi = 0.5$, $\eta = 0.5$. The plots show the time-series for each compartment. This choice of parameter values is such that, because the transitions are strongly correlated, individuals are only tested at the end of their infectious period and are effectively never isolated. The top row are the compartments representing unconfined individuals, those in $S_U$, $E_U$, $I_U$ and $R_U$. The bottom row are those representing isolated—diagnosed or distanced—individuals, $S_D$, $E_D$, $I_D$, $R_D$. The heavy orange curves are the output of the ODE-based simulation. The violet curves are the average output of the agent-based simulation with correlated transitions, with envelopes for one and two standard deviations.

Fig 8 shows simulations for the same scenario as Fig 6, but using the correlated ABM. In this case, as explained in the methods section, there is an important point to be made about what the testing rate represents. Here we have taken $\theta = 7^{-1}$days$^{-1}$ and $\theta_0 = 2\theta$, meaning that we assume the total rate means infectious individuals have a 50% likelihood of being tested 3.5 days after displaying symptoms. In an uncorrelated ABM or an ODE model, only the total testing rate $\theta$ matters and test events occur according to the underlying (exponential) distribution. In an ABM where events happen after a deterministic or strongly correlated time, this distinction matters. In particular, it's important to make sure that anyone who gets tested is tested soon enough in the course of their disease that they can be usefully isolated, before they've had time to spread it. If all testing happens exactly seven days after infection—the same length of time as the recovery period—testing will do nothing to prevent propagation of the disease. Given this consideration, we used a reasonable assumption that led to the same overall value of $\theta$ as the previous simulation. It should be noticed, however, that the ODE performs less well at matching this different model. The correlated ABM simulation results in more infections than the previous one. This shows the effect of transition time distributions; using constant rates, and thus exponential distributions, adds a significant component of very short-time transitions (both recoveries and testing/isolation) that actually end up improving outcomes. In weak testing regimes, where the waiting times can be quite long, this can cause an ODE model to make more optimistic predictions. Fig 9 instead shows what happens in a strong testing regime, where waiting times are short enough that the interval between infection and testing matters less. In this case, the ODE and ABM descriptions match quite well. The irregular appearance of the ABM curves here is due to the discrete nature of its transitions, leading to strong and correlated fluctuations.

## Methods

We consider the problem of determining the effect of testing and contact tracing in a population, $P$, consisting of a set of indistinguishable individuals among whom a disease propagates.

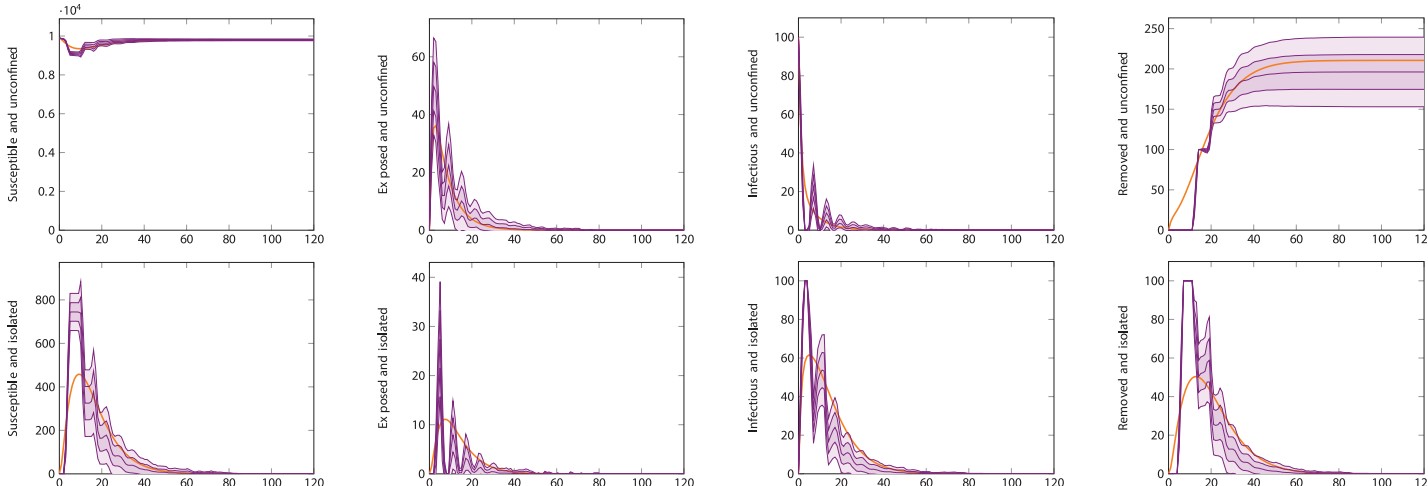

**Fig 9. Comparison of differential equation and agent-based model with correlated transitions.** As in Fig 8, the population size is 10000 individuals with 100 initially infected. The testing rate is $\theta = 2.5^{-1}$days$^{-1}$, the base testing rate is $\theta_0 = \theta$, and tracing rate and success probability are $\chi = 0.5$, $\eta = 0.5$. This choice of parameters exhibits a good match between the models because the testing rate, though strongly correlated, is faster than the recovery rate. The plots show the time-series for each compartment. The top row are the compartments representing unconfined individuals, those in $S_U$, $E_U$, $I_U$ and $R_U$. The bottom row are those representing isolated—diagnosed or distanced—individuals, $S_D$, $E_D$, $I_D$, $R_D$. The heavy orange curves are the output of the ODE-based simulation. The violet curves are the average output of the agent-based simulation with correlated transitions, with envelopes for one and two standard deviations.

To answer this we adapt the standard Susceptible-Exposed-Infectious-Removed (SEIR) compartmental model [22, 58] to incorporate contact tracing as well as testing and isolation of cohorts of people. Our adaptation extends the classic SEIR to not only include progression through disease stages from exposure, via infection to recovery, but to also keeping track of the changing make up of the population as the disease progresses. To achieve this we require our model to have two additional features:

1. to keep track of whether people have been isolated from the rest (either due to testing positive, or having been traced as a contact of someone who tested positive)

2. to keep track of whether people have been in contact with an infectious individual recently enough to be potential targets for tracing.

Ordinary compartment models like SEIR are designed to separate individuals into distinct, non-overlapping groups. This is not a problem for the first feature, as people who are isolated and people who are not constitute entirely distinct sets. We therefore can represent unconfined and isolated individuals simply by doubling the number of states, labeling $S_U$, $E_U$, $I_U$ and $R_U$ the Undiagnosed people who are respectively Susceptible, Exposed, Infectious, or Removed, and similarly, $S_D$, $E_D$, $I_D$ and $R_D$ the ones who have been Diagnosed or otherwise Distanced from the rest of the population, by means of home isolation, quarantine, hospitalisation and such.

However, dealing with contact tracing is harder, as it can not be achieved with separate compartments. Here we take two approaches. First, we describe an agent-based model that simulates contact tracing with an approximation of how it could take place in real life. This agent-based model serves as our reference. Then we describe fully our compartment model, and, relying on a system of second order Ordinary Differential Equations (ODEs), we introduce the concept of overlapping compartments. Overlapping compartments represent model states that are not mutually exclusive, so that it is possible for an individual to belong in more than one of them e.g. be infected and contact-traced, or exposed and tested. We define

equations for this model in order to represent the processes that happen in the agent-based model, providing the comparisons seen above in the Results section.

## An agent based model of contact tracing

Among the possible measures to suppress an epidemic, contact tracing is defined as "an extreme form of targeted control, where the potential next-generation cases are the primary focus" [67]. In other words, contact tracing is the process by which we aim to identify and isolate individuals who have been in contact with an infectious patient in the past and are thus more likely to have been exposed to the disease, in order to remove them from the pool of possible infectious patients before they develop symptoms.

We start by defining our modified SEIR model in agent-based form. The model features $N$ agents each characterised by a state symbolising progression throughout the disease ($S$, $E$, $I$, or $R$) as well as a single bit characterising whether they are Undiagnosed or Diagnosed/Distanced ($U$ or $D$). As mentioned above, we label $S_U$, $S_D$, $E_U$, etc. respectively the numbers of individuals in each combination of those states, and $S$, $E$, $I$, $R$ the totals ($U$ and $D$ combined). In addition, we store a contact matrix keeping track of which individuals have been in contact with which infectious members of the population, and an array of all those individuals for whom one past infectious contact has been identified, and thus they can be traced as potentially exposed individuals. We call $C_T$ the total number of such traceable individuals. This contact matrix encapsulates a history of interactions in a way that is realistic but is not possible to represent directly in ODE form. It is specifically the functioning of this individual contact matrix that we claim to reproduce at the population level with our ODE formulation below.

We simulate the model using Gillespie's algorithm [68], which provides a way to sample exact trajectories produced by such stochastic processes. The possible state transitions that can take place are:

1. contact between a random individual and one belonging to $I_U$, with rate $cI_U$. The contact is stored in the contact matrix. If the individual happens to belong in $S_U$, with likelihood $\hat{\beta} \leq 1$, the contact results in exposure, and the $S_U$ individual becomes $E_U$;

2. progression of the disease for an $E$ individual into $I$, with rate $\alpha E$;

3. recovery from the disease, or removal due to hospitalisation or death, for an $I$ individual into $R$, with rate $\gamma I$;

4. diagnosis by regular testing of an $I_U$ individual, with rate $\theta I$. The individual is moved to $I_D$; all its past contacts, retrieved from the contact matrix, are marked as traceable with likelihood $\eta \leq 1$. If the individual moved to $I_D$ was marked as traceable, it is unmarked (as they're already in isolation and there is no need to trace them any more);

5. release from isolation of an $S_D$ individual, making them $S_U$, with rate $\kappa S_D$;

6. release from isolation of an $R_D$ individual, making them $R_U$, with rate $\kappa R_D$;

7. contact tracing of a traceable individual with rate $\chi C_T$. The individual is moved from $X_U$ to $X_D$, where $X$ is whatever state of progression they are in, and they're removed from the list of traceable individuals.

The transitions described above can be intuitively seen as corresponding to the ones that would happen in an idealised real-life version of epidemic spread with testing and contact tracing. The biggest deviation from reality is the perfect mixing of the population implied by the first process. The testing and tracing processes are parametrised by $\theta$, the rate of diagnosis of

infectious individuals, $\eta$, the likelihood or efficiency with which the tracing process identifies contacts, and $\chi$, the rate at which they are found and isolated. We will describe the meaning and importance of these numbers as we explain how they fit into an ODE model description of the same processes.

## A time-correlated agent based model

We also define a second ABM, for the purpose of investigating how time correlation between events affects the results. In regular ODE models, and in the ABM that was described above, transitions between states happen at fixed rates but are completely uncorrelated; this results in an exponential distribution of times each agent spends in a given state. In real life, this is obviously an unrealistic scenario: in particular, the time necessary for someone to recover from a disease is generally better described by a peaked distribution [62–66]. A very simplified approach was taken here to approximate this situation. A second ABM model was developed, with the same compartments and transitions as the one described above, but with one key difference: all events that happened randomly now happen deterministically at a fixed time. This can also be seen as the times following a pure Dirac delta distribution. In order to enable a comparison, the time was chosen to be equal to the average time of transition for the uncorrelated model. This means for example the transition $I \to R$ at a rate proportional to $\gamma$ corresponds in this model to each individual in $I$ moving to $R$ precisely after an interval of $\gamma^{-1}$. Similarly, contacts between individuals happen regularly at intervals of $c^{-1}$.

The main difference between this model and the previous one is in the mechanism used to describe testing. In the regular ABM and in the ODE model, a single parameter $\theta$ is sufficient to describe the rate of testing. This includes both the speed at which an individual suspected of being infected can be tested and the probability of them being identified and tested at all. However, these two things are not the same for the purposes of a deterministic model. For example, a value of $\theta^{-1} = 14$ days might mean that everyone who is infected gets tested 14 days after infection, or that 50% get infected 7 days after, and the rest not at all. These can lead to very different outcomes in this model; in particular, if $\theta^{-1} > \gamma^{-1}$, no one will be tested before recovering, and thus, testing is as good as non-existent. For this reason, for this specific model, we further split the parameter in two:

$$\theta = f_\theta \theta_0 \tag{2}$$

with $\theta_0$ being the 'base' testing rate and $0 \le f_\theta \le 1$ the fraction of individuals in the $I$ state who are tested. In practice, in the software used for the simulation, $\theta$ and $\theta_0$ are defined as input parameters and $f_\theta$ is derived from their ratio. The waiting time for a test will then be $\theta_0^{-1}$ for a fraction $f_\theta$ of agents, infinite for everyone else.

## The standard SEIR model

We begin by introducing the ODE form of the standard SEIR model [22, 58]. Because of the large number of model compartments and exchange terms between them that will be featured in the full model, we introduce a systematic notation to refer to rates that link them. We refer to $\Delta_{X \to Y}$ as the rate at which members of the population move from compartment $X$ to compartment $Y$. For example, $\Delta_{S \to E}$ is the rate at which Susceptible members of the population are Exposed to the virus. In addition, for convenience when discussing movements that can happen due to multiple phenomena, we might add a superscript, such as $\Delta_{X \to Y}^{Z}$, to indicate only the part of that rate that can be ascribed to a given process $Z$.

With this notation, the differential equations that describe the standard SEIR model have the following form,

$$\frac{dS_U}{dt} = -\Delta_{S_U \to E_U} \tag{3}$$

$$\frac{dE_U}{dt} = \Delta_{S_U \to E_U} - \Delta_{E_U \to I_U} \tag{4}$$

$$\frac{dI_U}{dt} = \Delta_{E_U \to I_U} - \Delta_{I_U \to R_U} \tag{5}$$

$$\frac{dR_U}{dt} = \Delta_{I_U \to R_U} \tag{6}$$

Note that all terms involve compartments identified with U subscripts as these equations all apply to the undiagnosed part of our model. They will then be expanded upon to include the effects of isolation and testing in the next section.

The terms in the above differential equations are defined in the usual way as,

$$\Delta_{S_U \to E_U} = \beta S_U \frac{I_U}{N} \tag{7}$$

$$\Delta_{E_U \to I_U} = \alpha E_U \tag{8}$$

$$\Delta_{I_U \to R_U} = \gamma I_U \tag{9}$$

where $\beta = \hat{\beta}c$ is the infection rate, $\alpha$ is the disease progression rate and $\gamma$ is the disease recovery rate.

While this formulation treats the populations as continuous analytical functions, in general these equations describe the mean trajectory of what is fundamentally a stochastic system. This stochastic system can be simulated with Gillespie's algorithm and, up to this point, is equivalent in the continuous limit to an agent-based model featuring the same compartments and transition rates.

## The SEIR model with diagnosis and isolation

Now we add diagnosis to our description. Four more compartments, $S_D$, $E_D$, $I_D$ and $R_D$, are created to keep track of population cohorts who have been identified as potentially infected, and thus isolated from the rest of the population as a measure to limit the spread of the disease. Disease progression is not affected by this process; therefore,

$$\Delta_{E_D \to I_D} = \alpha E_D \tag{10}$$

$$\Delta_{I_D \to R_D} = \gamma I_D \tag{11}$$

Including isolation will change the infection rate, as unlike population $I_U$, the isolated population $I_D$ does not contribute to further infection. Hence we do not include an infection term here. This is an idealisation. In reality isolation will not be perfect, and we can imagine a reduced 'cross-infection' rate in which some people belonging to $S_U$ are infected by people in $I_D$. This could happen with medical professionals treating infectious patients or care workers who maintain a quarantine facility. We could even consider infection of people in $S_D$ due to

those in $I_D$, such as a patient in home isolation infecting their family. However, for present purposes, we will work in an ideal situation where isolation is perfect.

Finally, we need to incorporate mechanisms to move individuals between the $U$ and $D$ branches of the model. For this purpose we define a testing rate, $\theta$, which represents the fraction of people belonging in $I_U$ who, each day, are diagnosed with the disease. We note that this parameter does not refer to any specific testing procedure; it just represents the total of people who are recognised as having the disease. It can represent, for example, actual testing for a specific pathogen as well as clinical diagnosis. We only focus on the category of $I_U$ as these are the patients who are most likely to realise they are sick and seek medical help. This generic testing process is described by the equation,

$$\Delta^T_{I_U \to I_D} = \theta I_U \tag{12}$$

In addition, people will be released from isolation after a finite time without symptoms. For this reason, we don't include a mechanism for people in $I_D$ to return to the $U$ branch of the model, as they're likely to be symptomatic or test positive for the pathogen. Instead, we consider that people who have been isolated despite being not infected, or who are still isolated after having recovered, will return to normal conditions at a rate $\kappa$,

$$\Delta_{S_D \to S_U} = \kappa S_D \tag{13}$$

$$\Delta_{R_D \to R_U} = \kappa R_D \tag{14}$$

With this model adaptation, a single infected individual can now take two paths:

1. $S_U \to E_U \to I_U \to R_U$, in which they are exposed to the disease, become infectious, and finally recover, without being isolated or diagnosed, as in the normal SEIR model, or,

2. $S_U \to E_U \to I_U \to I_D \to R_D \to R_U$, in which, after becoming infectious, they are identified, isolated, removed from the pool of those who can infect other susceptible people, and after recovering, released from isolation.

Having these two paths allows attainment of some degree of control of the epidemic; however, it must be noted that while we have introduced them, the states $S_D$ and $E_D$ are here left unused. This is because at this stage we associate testing with symptomaticity; there is yet no mechanism other than by diagnosis to identify someone who could be infected. This is especially problematic in terms of the impossibility of isolating exposed people. These are individuals with a latent infection who will soon become infectious. Isolating them pre-emptively would contribute a great deal towards suppressing the epidemic. For this reason, we move on to include contact tracing as a means of preventive isolation.

## The SEIR model with testing, tracing and isolation

We've seen previously that it is intuitive how contact tracing can be represented in an agent-based model, in which individuals are simulated and each has an history of contacts with other members of the population. It is not as obvious how to treat contact tracing in a compartment model, where there is no memory of the histories of contacts of specific individuals, but only average quantities. We outline here a probabilistic method for doing this.

Let us define $\Pr(X)$ the probability of an individual of belonging to compartment $X$ of the population. For example, $\Pr(S_U) = S_U/N$ is the probability of an individual to be Susceptible and Undiagnosed. In addition, let us define $\Pr(C_I)$ the probability of an individual of having had contact with an infectious individual in the past where that infectious individual is still

infectious. The latter detail is important because here we consider only "next-generation" tracing; in other words, we only try to trace the direct contacts of those infectious individuals who were found to test positive. This is a conservative assumption. It could be possible to make contact tracing more effective by also tracing one generation further (the contacts of the contacts), but because the process requires exponentially more resources with each generation with decreasing likelihood of correctly identifying exposed or infectious individuals, we simply opt to neglect that possibility. Therefore, in this model the only people who can be traced are those whose most recent infectious contact is *still* infectious; once they recover, they cannot be identified as infectious any more, and thus it will be impossible to trace their contacts as well. Finally, we define $\Pr(C_T)$ the probability of an individual of being traced. All these probabilities are functions of time, and quantities that evolve with the model itself.

First, we rewrite the probability of being traced is

$$\Pr(C_T) = \Pr(C_T|C_I)\Pr(C_I) + \Pr(C_T|\neg C_I)\Pr(\neg C_I) \tag{15}$$

where $\Pr(C_T|C_I)$ is the conditional probability of being traced given that one has had an infectious contact in the past, and $\Pr(C_T|\neg C_I)$ the probability of being traced given that one has not. Clearly, $\Pr(\neg C_I) = 1 - \Pr(C_I)$. If we ignore the possibility of false positives, then $\Pr(C_T|\neg C_I) = 0$, namely, a person can only be traced if they did have an infectious contact in the past. If we then set an 'efficiency' parameter $\eta$ representing the fraction of contacts that we are indeed able to identify, the probability of being traced at a given time is simply

$$\Pr(C_T) = \eta\Pr(C_I) \tag{16}$$

To derive transition rates among compartments, we consider that individuals will be traced proportionally to how quickly the infectious individuals who originally infected them are, themselves, identified. We add a factor $\chi$ to account for the speed of the tracing process itself, and we find a global tracing rate,

$$\Delta^{(C_T)} = \chi\theta\Pr(C_T) = \chi\theta\eta\Pr(C_I) \tag{17}$$

It then follows that, for individuals in a given compartment $X$, the rate at which they're isolated by contact tracing is

$$\Delta^{(C_T)}_{X_U \to X_D} = \eta\,\theta\chi\Pr(X_U|C_I)\Pr(C_I)N = \eta\,\theta\chi\Pr(C_I|X_U)X_U \tag{18}$$

where in the last step we made use of Bayes' theorem [69]. This is our Eq 1, the central mathematical result of this paper.

The difficulty is then computing the exact probabilities. These are functions that, in general, vary in time and require a certain degree of information about the past. We need to define useful assumptions and approximations in order to work with these probabilities in a model that inherently lacks any memory about the individual histories of the elements of its population.

One simple assumption for Exposed and Infectious individuals is

$$\Pr(C_I|E_U) = \Pr(C_I|I_U) = 1 \tag{19}$$

meaning that we assume that if an individual has been Exposed or Infected, they must also have had an infectious contact in the recent past. This is in fact the reason why contact tracing is an effective use of resources: it skews heavily towards identifying those who have in fact been exposed to the disease. We remark that this assumption does not hold in general in circumstances where it is possible for an individual to become infected indirectly, such as by contact with contaminated surfaces. For present purposes we assume that the likelihood of such

events is small compared with the likelihood of being infected through contact with another individual.

Another limit of this assumption is that we have defined $\Pr(C_I)$ as the probability of having had an infectious contact *who is still infectious*. For $\alpha \ll \gamma$, or for some infectious individuals who may take a long time to recover, their original infector might have already recovered in the time it takes for them to be tested. However, here we study a model in which $\alpha > \gamma$, and it is reasonable to assume that those infectious individuals who are tested are identified relatively early on in their infection, especially if $\theta > \gamma$. Therefore, we deem the assumption in Eq 19 acceptable at least insofar as these two conditions hold and indirect infection is unlikely.

Estimating $\Pr(C_I|S_U)$ and $\Pr(C_I|R_U)$ is more complicated. One possible approximation is to work as if $I_U$ were constant on the time-scales of interest; in that case we would have

$$\Pr(C_I|S_U) = (1 - \beta)c\gamma'^{-1}I_U \tag{20}$$

$$\Pr(C_I|R_U) = c\gamma'^{-1}I_U \tag{21}$$

where $\gamma'$ is the overall rate at which individuals are removed from the $I_U$ state. Putting together recovery, regular testing, and contact tracing, we find $\gamma' = \gamma + \theta(1 + \eta\chi)$. The main difference between the two equations is determined by the fact that someone in $S_U$ might still be infected, and thus only has a probability $1 - \beta$ of remaining susceptible after a contact with an infectious member of the population, whereas for recovered individuals this is not an issue any more. Eqs 20 and 21 can be used to compute rates of contact tracing by combining them with 1. However, here we try to go beyond the crude approximation of constant $I_U$, as it may often reflect reality very poorly.

We consider for example the total number of members of $S_U$ who also have had recent infectious contacts, $N(C_I|S_U) = \Pr(C_I|S_U)S_U$. We can describe these in first approximation as

$$N(C_I|S_U)(t) = \int_{-\infty}^{0} (1 - \beta)cI_U(t - \tau)S_U(t - \tau)F_I(t, \tau)F_{S_U}(t, \tau)d\tau \tag{22}$$

where the $F_X(t, \tau)$ are the 'survival functions' for the state $X$. In other words, these are the functions that determine how likely it is that an individual that was in $X$ at time $\tau$ still is in the same state at time $t$. We also used $F_I$, meaning the survival function of the total number of infectious individuals, $I = I_U + I_D$, because here we focus on overall infectiousness, not the fact that one might have been isolated before recovery. Note, however, that only $I_U$ individuals participate in contacts. The reason that this is an approximation is that we're not excluding the $N(C_I|S_U)$ from the pool of $S_U$ that can be contacted, and thus there is a risk of double counting. That risk will remain negligible as long as $N(C_I|S_U)/S_U$ is small; therefore, this model will perform better in a regime in which there are few infectious individuals, and thus, few contacts. This is in fact the regime in which contact tracing is most likely to be feasible in practice, to control small outbreaks rather than in presence of an uncontrolled epidemic. Regardless, we show in the Results section that even when this approximation does not hold, while it results in oscillatory behaviour early on, it still generally adequately describes the overall trends and long term equilibrium. Eq 22 is equivalent to the integral form of an equation for a compartment model [70]. It can be written in differential form as,

$$\left.\frac{dN(C_I|S_U)}{dt}\right|_t = (1 - \beta)cI_U(t - \tau)S_U(t - \tau) - (h_I(t) + h_{S_U}(t))N(C_I|S_U) \tag{23}$$

where the $h_X = \frac{1}{F_X}\frac{dF_X}{dt}$ are the 'hazard functions' for the state $X$. In particular, $h_I = \gamma$.

Given the similarities between these equations and the ones describing the compartment models, it is natural to think of creating a specific compartment for $N(C_I|S_U)$. This is in fact what we do. There is, however, an important difference from regular compartments, because this compartment does not include individuals that exclusively belong to it; rather, it overlaps with $S_U$. It is more of a device used for book-keeping purposes, to compute the integral in Eq 22 within the confines of the model, than a compartment in the usual sense. We similarly define $N(C_I|E_U)$, $N(C_I|I_U)$ and $N(C_I|R_U)$, which leads, using Eq 1, to the following contact tracing rates,

$$\Delta_{S_U \to S_D}^{(C_T)} = \eta \, \theta \chi N(C_I|S_U) \tag{24}$$

$$\Delta_{E_U \to E_D}^{(C_T)} = \eta \, \theta \chi E_U \tag{25}$$

$$\Delta_{I_U \to I_D}^{(C_T)} = \eta \, \theta \chi I_U \tag{26}$$

$$\Delta_{R_U \to R_D}^{(C_T)} = \eta \, \theta \chi N(C_I|R_U) \tag{27}$$

In addition, we establish the following transition rates between these $N$ compartments,

$$\Delta_{\to N(C_I|S_U)} = (1 - \beta)cI_U S_U \tag{28}$$

$$\Delta_{N(C_I|S_U)\to} = (\gamma + \eta \, \theta \chi)N(C_I|S_U) \tag{29}$$

$$\Delta_{N(C_I|S_U)\to N(C_I|E_U)} = \beta cI_U N(C_I|S_U) \tag{30}$$

$$\Delta_{\to N(C_I|E_U)} = cI_U E_U + \beta cI_U S_U \tag{31}$$

$$\Delta_{N(C_I|E_U)\to} = (\gamma + \eta \, \theta \chi)N(C_I|E_U) \tag{32}$$

$$\Delta_{N(C_I|E_U)\to N(C_I|I_U)} = \alpha N(C_I|E_U) \tag{33}$$

$$\Delta_{\to N(C_I|I_U)} = cI_U^2 \tag{34}$$

$$\Delta_{N(C_I|I_U)\to} = (\gamma + \theta + \eta \, \theta \chi)N(C_I|I_U) \tag{35}$$

$$\Delta_{N(C_I|I_U)\to N(C_I|R_U)} = \gamma N(C_I|I_U) \tag{36}$$

$$\Delta_{\to N(C_I|R_U)} = cI_U R_U \tag{37}$$

$$\Delta_{N(C_I|R_U)\to} = (\gamma + \eta \, \theta \chi)N(C_I|R_U) \tag{38}$$

There is a lot going on in Eqs 28–38; most importantly, these new compartments do not conserve the total size of the population. Their membership grows as contacts happen and shrinks as time passes. All the key processes can be summed up as follows:

- elements are 'created' for each state proportionally to the rate of contact with individuals belonging to $I_U$, adjusted with $1 - \beta$ in the case of $S_U$ to account for the likelihood that the

contact is infective. These terms are 'sources' and can be recognised by having an arrow with nothing on its left in the subscripts;

- elements 'decay' at a rate that amounts to $\gamma$ (the hazard function for $I$, which always appears as it refers to the original infector) plus a rate representing the hazard function for the transition $X_U \rightarrow X_D$. These terms are 'sinks' and can be recognised by having an arrow with nothing on its right in the subscripts;

- elements move between compartments following the usual transitions that control the dynamics of the SEIR model (infection, progression of the disease, recovery). These terms are analogous to the corresponding ones connecting $X_U$ states, and contribute the remainder of the hazard function for each $X_U$ to Eq 23 and equivalents.

It must also be noted that, in practice, considering Eq 19, it must be $N(C_I|E_U) = E_U$ and $N(C_I|I_U) = I_U$, which removes the need for two of the four compartments above and simplifies the equations to

$$\Delta_{\rightarrow N(C_I|S_U)} = (1 - \beta)cI_U S_U \tag{39}$$

$$\Delta_{N(C_I|S_U)\rightarrow} = (\gamma + \eta\,\theta\chi + \beta cI_U)N(C_I|S_U) \tag{40}$$

$$\Delta_{\rightarrow N(C_I|R_U)} = cI_U R_U + \gamma I_U \tag{41}$$

$$\Delta_{N(C_I|R_U)\rightarrow} = (\gamma + \eta\,\theta\chi)N(C_I|R_U) \tag{42}$$

A few words are necessary on the hazard function for the $X_U \rightarrow X_D$ transitions. This is approximated as $\eta\theta\chi$ in states $S_U$ and $R_U$ even though that is not precisely correct; the correct hazard function would be $\eta\theta\chi N(C_I|X_U)/X_U$, but that introduces a risk of instability for small values of $X_U$. We justify this choice by the following reasoning. In a weak testing regime ($\eta\theta\chi \ll \gamma$), $N(C_I|X_U)/X_U$ might be high due to a great number of infected individuals, but in principle should never be greater than 1 (modulo the point above about double counting). Therefore, the hazard function is dominated by $\gamma$. Conversely, in a strong testing regime, the number of infected individuals, and thus $N(C_I|X_U)/X_U$, will be very small, and this assumption will at most end up underestimating the effect of contact tracing (by causing a faster decay in $N(C_I| X_U)$ than otherwise would happen). The examples shown in the Results section illustrate how this affects the simulations—in general, leading to good predictions for the behaviour of the $E_U$ and $I_U$ compartments.

Eqs 7–9, 10, 11, 12, 24–27 and 28–38, together, define entirely our model. The parameters that appear in these equations are summarised for reference in Table 1.

**Table 1. Parameters used in the SEIR-TTI model.**

| Parameter | Description |
|:---:|:---|
| $N$ | Population size |
| $c$ | Average contacts per day |
| $\hat{\beta}$ | Transmission rate per contact |
| $\alpha^{-1}$ | Incubation period (time from exposed to infectious) |
| $\gamma^{-1}$ | Recovery period (time from infection to recovery) |
| $\theta$ | Testing rate of infectious individuals |
| $\eta$ | Efficiency or success rate of contact tracing |
| $\chi$ | Contact tracing rate |

### Software implementation

We implement the above ordinary differential equations and agent-based model in our PTTI Python package (https://github.com/ptti/ptti) using the Compyrtment [71] package that facilitates the formulation of initial value problems. It is written for Python 3 and makes use of the scientific computation libraries NumPy and SciPy [72, 73] as well as the optimisation library Numba [74]. The specific scripts used to run the simulations and produce the figures seen in this paper can be found in the `ptti-theory-paper` branch of the repository.

The PTTI package provides a declarative language for specifying simulations of models implemented as Python objects. It supports setting of model parameters, simulation hyperparameters as well as interventions that modify parameters at particular times to conduct piece-wise simulations reflecting changing conditions in a convenient and user-friendly way. We hope that this software formulation will be useful for easy and rapid exploration of the effects of different intervention scenarios for disease outbreak control.

## Discussion

Our work outlines a method for extending the classic SEIR model to include Testing, contact-Tracing and Isolation (TTI) strategies. We show that our novel SEIR-TTI model can accurately approximate the behaviour of agent-based models at far less computational cost. Our adaptation is applicable across compartmental models (e.g. SIR, SIS etc) and across infectious diseases. We suggest that the SEIR-TTI model can be applied to the COVID-19 pandemic to understand the impact of possible TTI strategy to control this outbreak.

The importance of modelling to support decision making is widely acknowledged, but models are far more useful when they can accurately represent the classes of interventions that are being considered [20]. The approach described in this paper enables accurate and efficient modelling of contact tracing and testing across a wide range of relevant parameter values. The ability to accurately model TTI strategies across parameter values is vital for controlling disease outbreaks including the current COVID-19 pandemic. Effective testing, contact tracing and isolation strategies have been the key measures that have prevented the epidemic spreading in South Korea [75], New Zealand and Germany [76].

Our work is novel as it is to date, and to the best of our knowledge, the first deterministic model to explicitly incorporate contact tracing. Previously, an attempt to model contract tracing was made by Fraser et al. [57]. The model was based on the McKendrick-Von Foerster [77] partial-differential equation that describes dynamical systems in terms of time and one more independent variable and which can be integrated along the characteristic lines (method of characteristics) to produce a system of ordinary differential equations analogous to the SIR system. The McKendrick-Von Foerster equation in [57] described dynamics of the current population at time t as a function of those infected some time ago (current time $t$ and previous time $\tau$ are the two independent variables in [57]). This equation was also studied more recently in the context of the French COVID-19 epidemic where the two independent variables were time and age in [78]. Fraser et al. [57] modelled contact tracing and isolation as two independent processes determined by the same distribution and individuals that are infectious were subdivided into four groups of individuals: those individuals who will never be isolated or contact-traced; individuals who will be isolated but never contact-traced; individuals who will never be isolated but will be contact-traced and individuals who will be either isolated or contact-traced. The main assumptions of the model are the two probabilities: firstly the probability with which individuals become symptomatic and isolated; then once symptomatic individuals who have been isolated have their contacts traced, the people they have infected are themselves quarantined with some second probability. Level of contact tracing is not a specific parameter

in this model. Furthermore, contacts of individuals who are asymptomatic when quarantined are only themselves traced after symptoms develop.

Unlike this model, we have explicitly incorporated in our framework tracing level of both exposed and infectious people—hence allowing the pool of traced people to be increased and specifically accounting for the two groups. Furthermore, we also consider that those traced will be isolated with certain probability and hence we view isolation as follow-on process from tracing and dependent on it. The main purpose of the model in [57] is show that the proportion of transmission that occurs before symptom occur i.e. the proportion of asymptomatic infection ($\theta$ in their model) is a useful new statistic for describing whether isolation- or contact-tracing-based intervention measures are better at controlling an epidemic outbreak. Their results suggest that only if asymptomatic infection is above a certain threshold ($\theta > 1/R_e(0)$) contact tracing needs to be added to the set of control measures. But the issue with an emerging pandemic, such as COVID-19, is that we do not know the proportion of asymptomatic infection $\theta$. Our model instead, allows contact tracing level to be included from the onset of an emerging pandemic and to be varied for both exposed and infectious people. Importantly, we aim to quantify how the interplay between testing and tracing is important in controlling outbreaks –it is the balance between these that quantifies how effective reproduction number changes—as illustrated in Fig 5.

Contact-tracing has been until now typically modelled successfully with agent-based models. We are aware that agent-based models allow more realistic infectiousness profiles to be incorporated, and we have done so in our other work [12] as have other studies [79, 80]. We are also aware that ABMs allow more realistic distribution of times spent in each state and can incorporate fixed time delays for testing and tracing rather than constant rates which lead to exponential waiting time distributions.

An important aspect of our approach is that our ODE formulation *explains* the behaviour of anthe agent-based model.

Namely, agent-based models are formulated in terms of local interactions among individuals and exhibit emergent behaviour at the population level. For interesting agent-based models, it is usually difficult to obtain any explicit connection between the local interactions and the population-level dynamics except through simulation and inspection of the results. We argue that our work here shows such an explicit connection: we have been able to capture the dynamics that arise at a population level from testing and contact tracing. We show that this is correct by demonstrating good agreement with the population-level dynamics that emerge from the agent-based formulation where only local interactions are specified.

The SEIR-TTI model here considers disease propagation in the classical well-mixed setting. This is appropriate especially in circumstances where data are sparse and gives qualitatively similar results to those from fine-grained models that might otherwise provide more quantitatively accurate results if only more detailed data were available. In particular, well-mixed models do not include any notion of the network of contacts across which a contagion spreads in the real world. In reality, individuals in a large population are not equally likely to have contact with one another and it has long been known [48–50, 52, 53, 81–83] that heterogeneity in underlying population structure can have a strong effect [42, 84–86] on disease propagation. This effect is distinct from the choice of distributions reflecting the natural history of the disease: whereas peaked distributions are appropriate for transitions between states caused by the progression of the illness, the distribution of infection events mediated by a contact network are very different. Both of these classes of distribution are different from the exponential distribution implied by the underlying mass action semantics of a well-mixed model. Future work will include developing a better understanding of the relationship between network structure

and effectiveness of tracing, and mathematical characterisation of the classes of solution available for these models.

Another extension is investigating the extent to which individual decisions about compliance with measures to reduce disease propagation (voluntary distancing, wearing of masks, etc.) affect the success of containment. A game-theoretical approach such as that considered by Zhao et al. [87] may produce useful insights into this question. Insights gained from these extensions can inform policy design for relaxing onerous restrictions on the population.

An important next step in this work is the real-time policy driven application of SEIR-TTI. As our next piece of work we are planning to explore how SEIR-TTI model can be combined with economic analysis to guide decisions around optimal design of a TTI strategy that can suppress the COVID-19 epidemic in the UK.

## Conclusion

This paper shows how to extend compartmental models to incorporate testing, contact tracing and isolation. The resulting SEIR-TTI model is a key development in the widely used SEIR models, and an important step if these are to be useful in policy decision making during outbreaks. The long and successful history of testing, contact tracing and isolation in slowing and stopping the spread of infectious diseases is well known [67], with clear immediate importance for COVID-19 control [88].

The design of policies that include a variety of infectious disease control tools, and understanding and applying them in ways that are effective for society at large, is critical. Tools and models that allow policymakers to better understand the policies and the dynamics of a disease are therefore critical. If making policy decisions without evidence is flying blindly, making decisions without understanding the consequences of the various control measures is flying without flight controls. Models like SEIR-TTI can inform policymakers of the role that testing and tracing can play in preventing the spread of disease. Combined with economic and policy analysis, this can enable far better decision making both in the immediate future, and in the longer term. The next step in our work is indeed this: the application of the SEIR-TTI model combined with economic models to investigate the effect of different TTI strategies to conquer the COVID-19 epidemic in the UK.

## Author Contributions

**Conceptualization:** Simone Sturniolo, William Waites, Jasmina Panovska-Griffiths.

**Formal analysis:** Simone Sturniolo, William Waites, Jasmina Panovska-Griffiths.

**Investigation:** Simone Sturniolo, William Waites, Jasmina Panovska-Griffiths.

**Methodology:** Simone Sturniolo, William Waites, Jasmina Panovska-Griffiths.

**Resources:** Simone Sturniolo.

**Software:** Simone Sturniolo, William Waites.

**Supervision:** Jasmina Panovska-Griffiths.

**Validation:** Simone Sturniolo, William Waites.

**Writing – original draft:** Simone Sturniolo, William Waites, Jasmina Panovska-Griffiths.

**Writing – review & editing:** Simone Sturniolo, William Waites, Tim Colbourn, David Manheim, Jasmina Panovska-Griffiths.

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
