## [Decision Letter · Decision Letter 0]

1 Oct 2020

Dear Dr Panovska-Griffiths,

Thank you very much for submitting your manuscript "Testing, tracing and isolation in compartmental models" for consideration at PLOS Computational Biology. As with all papers reviewed by the journal, your manuscript was reviewed by members of the editorial board and by several independent reviewers. The reviewers appreciated the attention to an important topic. Based on the reviews, we are likely to accept this manuscript for publication, providing that you modify the manuscript according to the review recommendations.

We apologize for the delay in our decision. It was extremely challenging to find experts who had the capacity or the willingness to review your work. Unfortunately, we ended up with only one scientific review report. This report is, in our view, insightful and generally positive. Because we, the editors are also positive about your work we decided to take a decision with fewer reports than usual.

It would be great if you would address the points of criticism the reviewer raises, and that we believe are well taken. Especially the point on extending your perspective of ABM by more realistic infectivity profiles and distributions of time spent in each state would be worthwhile to discuss as a limitation. We also encourage you to include a comparison with an ABM that makes more realistic assumptions in this regard, if feasible.

In addition, there is previous work by Fraser et al that you mention in the Introduction (ref 52) that is very similar in scope to your work. It would strengthen your work, in our opinion, if you compared your work to the approach by Fraser et al in more detail in the Discussion.

Lastly, please also note the suggestions in the reproducibility report to improve documentation and reporting of initial conditions.

Sincerely,

Roland R Regoes

Associate Editor

PLOS Computational Biology

Virginia Pitzer

Deputy Editor

PLOS Computational Biology

[LINK]

Reviewer's Responses to Questions

**Comments to the Authors:**

Reviewer #1: The Reproducibility Report is uploaded as an attachment.

Reviewer #2: In this work, Sturniolo et al. formulate a method for modelling contact tracing in a compartmental model and test this ODE approximation against an ABM with the same structure. Overall, I think this work is valuable and should be published. I particularly liked the clear discussion of when the ODE approximation breaks down.

That said, I do have one major comment relating about the impact of constant infectivity and constant transition rates in ODE models. Ideally, this would require additional comparison between the ODE model and ABM. If this involves a prohibitive amount of work, it could potentially be adequately addressed through additional discussion.

I also have some additional points relating to presentation.

1. Exponential time distributions in ODE and ABM

The advantage of ABMs is discussed in terms of allowing fine-grained modelling of individual variation in susceptibility to disease and contact patterns. The authors point out that we often lack data to parametrise such fine grain models.

However, in the context of contact tracing, ABMs provide additional flexibility for important parameters we do have data about:

i) more realistic infectiousness profiles than the constant infectiousness assumed in ODE models (for covid infectiousness profiles, see for example He et al., 2020 Nature Medicine; Ferratti & Wymant et al, 2020 Science);

ii) more realistic distribution of times spent in each state – for example, fixed time delays for testing and tracing rather than constant rates (which lead to exponential waiting time distributions).

These are of course standard limitations of ODEs, but I think it may be particularly important when the aim is to model contact tracing: the effectiveness of tracing depends on where in the infectiousness profile individuals are isolated. The interplay between assumptions about the shape of the infectivity profile and waiting time distributions may therefore have a significant impact.

This manuscript would be considerable stronger if the authors included a comparison with an ABM with different assumptions of infectivity profile and time distributions (even for just one of the processes – e.g. the time it takes to trace contacts), to give a reader a sense of the magnitude and direction of the error this may introduce.

However, if implementing such a comparison is a prohibitive amount of work, discussing this point might be enough, particularly if the authors are able to provide an intuition for how these approximations might affect the output of the ODE.

2. Framing of testing and interpretation of theta

In the results section, testing and isolation (without tracing) is presented as random testing of the “entire population” (but also, confusingly, “only infectious individuals are tested and isolated”). Theta is therefore interpreted as the average testing rate in the population.

In the methods, the same process is described as infectious individuals experiencing symptoms and seeking diagnosis. Theta is therefore interpreted as the rate of experiencing symptoms and receiving a diagnosis.

Firstly, this inconsistency makes the paper confusing to read.

Secondly, it is not clear to me that that these two framings of theta are equivalent: if the entire population gets tested every 14 days on average, the expected time delay between getting infected and tested would be 7 days.

I think the authors are aware of this, based on the last paragraph on page 6 (or is theta = 1/14 a typo? The value of theta is 1/7 in subsequent figure legends), so it is not clear to me whether I should interpret the result that “testing the entire population every 20 days” as an expected delay from infection to diagnosis of 20 or 10 days.

3. Minor points:

- Calling the D compartment “diagnosed” is confusing when it also applies to susceptible individuals. In particular, this makes figure 1 difficult to interpret.

- Some variables are not defined when introduced (eta in equation 1, C_I in Figure 1).

- R stands for both recovered and reproductive number, which makes figures 2 and 4 a little confusing (from the axes labels, one would assume R(t) is recovered).

**Have all data underlying the figures and results presented in the manuscript been provided?**

Reviewer #1: None

Reviewer #2: Yes

PLOS authors have the option to publish the peer review history of their article (what does this mean?). If published, this will include your full peer review and any attached files.

Reviewer #1: **Yes: **Anand K. Rampadarath

Reviewer #2: No
---

## [Editor Report · Decision Letter 1]

14 Dec 2020

Dear Dr Panovska-Griffiths,

We are pleased to inform you that your manuscript 'Testing, tracing and isolation in compartmental models' has been provisionally accepted for publication in PLOS Computational Biology.

Best regards,

Roland R Regoes

Associate Editor

PLOS Computational Biology

Virginia Pitzer

Deputy Editor

PLOS Computational Biology

---

## [Editor Report · Acceptance letter]

4 Feb 2021

PCOMPBIOL-D-20-00860R1 

Testing, tracing and isolation in compartmental models

Dear Dr Panovska-Griffiths,

I am pleased to inform you that your manuscript has been formally accepted for publication in PLOS Computational Biology. Your manuscript is now with our production department and you will be notified of the publication date in due course.

With kind regards,

Alice Ellingham
